# Mobile sex-tech apps: How use differs across global areas of high and low gender equality

Amanda N. Gesselman[1]*, Anna Druet[2], Virginia J. Vitzthum[1,2,3]

**1** The Kinsey Institute, Indiana University, Bloomington, IN, United States of America, **2** Clue by BioWink GmbH, Berlin, Germany, **3** Department of Anthropology, Indiana University, Bloomington, IN, United States of America

* agesselm@indiana.edu

**Data Availability Statement:** All relevant data are within the paper and its Supporting Information files.

**Funding:** Clue by BioWink GmbH provided support for this study in the form of salaries for Authors VV

## Abstract

Digital technologies are increasingly intertwined into people's sexual lives, with growing scholarly interest in the intersection of sex and technology (sex–tech). However, much of the literature is limited by its over emphasis on negative outcomes and the predominance of work by and about North Americans, creating the impression that sex–tech is largely a Western phenomenon. Based on responses from 130,885 women in 191 countries, we assessed how women around the world interact with mobile technology for sex-related purposes, and whether in areas of greater gender inequality, technological accessibility may be empowering women with knowledge about sexuality. We investigated women's use of technology to find sexual partners, learn about sex and improve their sexual relationships, and track their own sexual health. About one-fifth reported using mobile apps to find sexual partners. This use varied by region: about one-third in Oceania, one-fourth in Europe and the Americas, and one-fifth in Asia and Africa. Staying connected when apart was the most commonly selected reason for app use with a sexual partner. About one-third had used an app to track their own sexual activity. Very few reported that the app they used to improve their sexual relationships was detrimental (0.2%) or not useful (0.6%). Women in countries with greater gender inequality were less likely to have used mobile apps to find a sexual partner, but nearly four times *more* likely to have engaged in sending and receiving sexts. To our knowledge, this study provides the most comprehensive global data on sex–tech use thus far, demonstrates significant regional variations in sex-tech use, and is the first to examine women's engagement in sex-related mobile technology in locations with greater gender disparities. These findings may inform large-scale targeted studies, interventions, and sex education to improve the lives of women around the world.

## Introduction

The pervasiveness of mobile Internet access is rapidly changing the way people navigate their lives including their romantic and sexual relationships. Social media platforms, online dating services, health and sex-related information, and apps to track virtually any experience and behavior are all readily accessible via one's mobile device. The increasing use of mobile sex–

and AD. The specific roles of these authors are articulated in the 'author contributions' section. The funders had no role in study design, data collection and analysis, decision to publish, or preparation of the manuscript.

**Competing interests:** The authors have read the journal's policy and the authors of this manuscript have the following competing interests: Authors VV and AD are paid employees of Clue by BioWink GmbH. There are no patents, products in development or marketed products to declare. This does not alter our adherence to PLOS ONE policies on sharing data and materials.

tech (defined by Gallop [1] as the use of innovative technology designed to enhance sexuality through knowledge, connections, archival data, etc.) has prompted scholars to devote greater attention to the impact of technology on sexuality and sexual behavior, and to the associated personal and relational consequences.

Despite this interest, much of the existing scholarship on sex-related use of technology is limited by its over-emphasis on the potentially negative outcomes of sex-tech use and by the narrow selection of study populations. Several studies and extensive media coverage have suggested that the use of mobile technology in our sexual lives comes at a heavy cost, with most attention directed at the consequences of this use in certain at-risk subgroups. For example, researchers have documented the adverse consequences of online dating apps on college students' binge drinking and on gay and bisexual men's risk for HIV (e.g., [2–5]). Understanding and mitigating any such risks for sex-tech users is paramount. But neglecting to evaluate the potentially positive impacts of sex-related uses of mobile technology arguably gives sex educators, researchers, and the general public a one-sided view of the impacts of this technology on sexual and relational health and well-being.

In addition, the great majority of research in this area has been produced by, and are studies of, North Americans, with a few additional studies in Western Europe (e.g., [6,7]) and China (e.g., [8,9]). This localization of research perpetuates the assumption that mobile sex–tech use is largely a Western phenomenon, leaving us ignorant of behavioral patterns in other areas of the world ("Western" in this paper refers collectively to Western Europe and North America, the latter comprising Canada and the USA).

Not surprisingly, romantic love and sexual desire have been documented in nearly all societies [10–12]. Additionally, smartphone access has rapidly spread around the world. A 2018 study conducted by the Pew Research Center showed that 76% of people in advanced economies (e.g., United States, United Kingdom, Australia, Israel, South Korea) and 45% of people in emerging economies (e.g., India, Indonesia, Kenya, Nigeria, Tunisia) have smartphones, although this tends to be skewed toward more ownership in younger populations [13]. With rising access to smartphones, and a near-universal motivation to seek romantic and sexual connection, mobile technologies that serve to enhance or advance these relationships are likely to have spread beyond those few regions that have been studied to date. On the other hand, countries and regions differ in their norms and practices, such as holding more conservative views regarding gender roles. While these factors may influence the use and impact of sex-tech, socio-cultural ideals do not necessarily prevent people from engaging in the proscribed behaviors. For example, foundational research by Alfred C. Kinsey revealed considerable same-sex sexual behavior, although these behaviors were illegal at the time [14,15]. Contemporary research has also demonstrated same-sex and premarital sexual behavior in countries where these behaviors are still illegal, such as in Saudi Arabia and Indonesia [16,17]. The nature and extent of such norms and practices on sex–tech behavior are, as yet, unanswered questions.

In this study, we contribute to addressing these significant gaps in current knowledge. We examine the global patterns of three sex-relevant uses of mobile technology and consider potentially positive as well as negative impacts of such use. In a non-representative sample of 130,885 women from 191 countries, we asked about their personal use of mobile sex–tech for meeting and connecting with sexual partners, for learning about sex and improving sexual relationships, and for tracking personal sexual health. Additionally, because women's sexuality and sexual behavior are likely to be affected by culturally shaped gender dynamics, we examined differences in the patterns of sex–tech use by the United Nations' country-specific Gender Inequality Index [18]. Through this examination, we hope to improve understanding of women's mobile sex–tech behaviors beyond a handful of (mostly Western) countries and to inform large-scale targeted studies, interventions, and sex education dissemination tools.

Our study was informed by the diffusion of innovation (DOI) theory [19], which provides a foundation for assessing how, when, and why new ideas and technologies spread and become adopted for long-term use. DOI has been especially useful in understanding the spread of mobile devices across the world, and in designing interventions focused on disbursing new sexual health information and practices within at-risk groups. For instance, in studying the uptake of mobile technology in China, Wei [20] discovered that a significant predictor and motivator—and thus a substantial driver behind the adoption of mobile devices—was the desire to be seen as in touch with Western trends. This allowed for better predictions of long-term use, and a better understanding of influential factors in the spread of ideas and attitudes toward new technologies. Additionally, DOI informed one of the first successful intervention strategies to reduce new HIV infections among those living in San Francisco [21] and has been used many times in similar research (e.g., [22,23]). More recently, researchers applied DOI to a study of a Ugandan intervention to observe the factors that influenced diffusion of new ideas and behaviors around intimate partner relationships and violence [24].

## Meeting and connecting with sexual partners

When first introduced in the U.S., online dating was stigmatized and users were perceived by others as socially incompetent, desperate, immature, and self-centered [25,26]. However, the majority of the U.S. population has now come to view online dating as a good way to meet people and find compatible partners [27,28]. As of 2016, over one in five U.S. adults aged 18 to 44 years had tried online dating, a marked increase since 2013, due largely to the greater popularity of smartphone dating apps over websites [29]. Evidence from Belgium [30], Germany [31], Hong Kong [32], the Netherlands [33], and Slovenia [34] suggests that the prevalence of online dating app use is growing, but there is little to no relevant data for most of the world.

Sexting is the transmission of sexual texts, pictures, or videos via mobile phone or other electronic media [35,36]. In a nationally representative sample of U.S. adults, 21% reported sending sexts and 28% reported receiving sexts [37]. Studies of adult sexting have reported negative correlates associated with sexting behavior, including risky sexual behavior, substance use, depression, and the unauthorized forwarding or sharing of sexts leading to humiliation, harassment, or abuse (e.g.,[37–43]). However, sexting between consenting adults may be beneficial to the individual and their relationship partners. For instance, sexting has been linked with increases in relationship satisfaction among those in committed relationships [44,45] and among those with insecure attachment styles [46]. In a recent study, researchers observed that about half of young adult sexters reported positive consequences associated with their sexting behaviors, including sexual and emotional relationship benefits [35]. Additionally, sexting may provide a safe space in which one can experiment with sexual desires and expression via production of sexual media, with anonymity if desired [47]. This may be particularly important for women in areas where it would be unsafe to engage in any form of outward sexual expression.

Collectively, these findings make clear the significant role of mobile-based technology for meeting, connecting, and communicating with potential and existing partners. While nearly all evidence has been produced by, and with samples of, U.S. adults, user statistics and some research in countries outside of the U.S. suggest that the interconnectedness of mobile technology and romantic and sexual relationships is not unique to North America. As the incorporation of mobile technology into our daily lives continues to expand, it is important to monitor this new development in intimate relationships to better understand the contemporary foundations of relationships and the resulting outcomes.

## Learning about sex and improving sexual relationships

The majority of adolescents and young adults use the internet to gather health information [48,49]. Notably, more of these searches concern sexual health over any other health topics [48]. In one study, the majority of millennial participants (e.g., persons born between 1981 and 1996) reported using the internet as a primary source of information about safe sex and sexual health, even though they believed medical professionals and schools to be the best source for such information [50]. Internet sources may be particularly important for people in locations or belonging to subgroups that are unlikely to receive relevant, tailored information. In the U. S., 78% of gay and lesbian youth and 65% of bisexual youth reported turning to the internet for sexual health information [51]. Mobile internet access via smartphones has also proved helpful to healthcare providers and educators seeking to connect with persons in poorer economic brackets or having less education, or with members of minority populations [52,53].

Seeking sexual information online indicates that the searcher has a need or desire for additional knowledge about a specific aspect of sexuality. The searcher may not have another venue for accurate and relevant sex education, or may be in an area or group in which talking about sex-related subjects is considered inappropriate or dangerous. Alternatively, these searchers may have received sex education and information from others, but this information may have been negatively valanced or fueled by a political or religious agenda, such as abstinence-only sex education. As such, online information-seekers may be turning to the internet to learn *positive*, or at least neutral, information about sex, empowering themselves to improve their sexual lives and relationships. In the current study, we collected information on whether women around the world have used internet-based technologies to learn about sex and improve their sexual relationships, as well as the specific relationship domains they intended to improve.

## Tracking personal sexual health

"Digital health" (also termed eHealth) encompasses the ways in which digital technologies are utilized in medicine and public health. These mobile technologies, including biosensors (e.g., FitBit) and a host of apps, promote the passive or active monitoring of one's own body or behavior to improve personal health and reduce healthcare costs [54–56]. These apps often provide a variety of medical and health information, and even detailed analysis of an individual's recorded data. These analyses may be compared with their own historical data or with data from other users. They may also be used to optimize goal-oriented health behaviors, create a health record, and monitor or manage health conditions or health outcomes.

A number of apps provide platforms for users to learn about and discuss sexuality, sexual behavior, menstruation and reproduction, and sex-related illnesses and conditions. Users can measure or record and input their own information, typically including their menstrual cycle information, sexual activities, histories, and/or symptoms. Many apps include information and tools related to the possible consequences of sex, including pregnancy (aiding in both avoidance and facilitation of conception), and tracking options for fertility biomarkers (e.g., basal body temperature, cervical fluid, and at-home ovulation test kit results). Additionally, some apps include information on identifying and tracking symptoms of sexually transmitted infections (STIs), such as irregular vaginal discharge. More niche apps also have the ability to track the duration of each sexual event if activated before the start of the event, number of "strokes" during the event, and the volume of sound produced during the event, to provide an overview of "performance" and a comparison to the performance of other users. In addition to providing the user with feedback about their own activities and health that may then lead them to make healthier decisions, researchers have also harnessed apps as tools for large-scale data

collection and public health outreach. For instance, many of these apps have sharing capabilities, in which the user can send their data to partners or friends, or perhaps to researchers conducting relevant studies. These apps also provide an easily accessible and user-friendly medium for healthcare delivery, including STI prevention, notifying partners of possible sexually transmitted infections, and sexual health promotion and education [57–60].

Beyond a few qualitative studies of relevant apps that are currently available (e.g., [61]), to our knowledge there are no investigations of women's use of health-tracking apps in relation to their own sexual behavior and experiences. Self-monitoring of this type indicates that the recorded behaviors and experiences are meaningful to the user, so much so that they feel it is important to create a digital history. Understanding what information is being tracked by women globally gives insight into which issues and aspects of their sexual lives these women deem important to archive, and can inform researchers of the regions in which women are or are not actively recording their sexual behaviors and experiences, so as to inform future research, education, and outreach.

### Current study

In the current study, we first examined whether women have used an online dating app to find a partner, and if so, the types of partners they were seeking. We also asked whether they had engaged in sexting, and if so, what mediums they had use for their messages (i.e., audio, video, text). Second, we asked whether these women had used mobile technology to learn about sex or to improve their sexual relationships, and which aspects of the relationship they thought had been improved by this use, or if this use had been unhelpful and even detrimental. Third, we asked whether women tracked aspects of their own sexual health. In the following sections, we present these data on multiple levels. We first provide overall descriptive statistics for each of the items, followed by a breakdown by global major regions and sub-regions (as defined by the United Nations) for a more focused view of how our participants' sex–tech behaviors differed by region. Last, because a focus on women's sexuality necessarily requires considering gender dynamics, we evaluated whether sex–tech behavior had any association with country-specific gender inequality using the Gender Inequality Index developed by the United Nations [18].

## Materials and methods

### Procedure

Data were collected during June 2017 via an anonymous online questionnaire created by the femtech company, Biowink GmbH, the developers of the Clue period and health tracker app, with consultation from the collaborating authors of this paper. The questionnaire was translated from English into nine other languages (French, Danish, German, Spanish, Portuguese, Italian, Simplified Chinese, Japanese, and Russian) and disseminated using Clue's newsletter, website, and social media accounts, and the social media accounts of the Kinsey Institute. All data were collected by Clue researchers. No personal identifiers (e.g., name, email address, IP address, browser or phone unique ID, Clue account identifiers, browser cookies) were collected. De-identified data were shared with the collaborating authors on this paper, and with no one else. The full survey and data file can be found in S1 Data and S1 Survey Tool, respectively.

### Participant's attributes

The sample comprised 130,885 women from 191 countries, 85.7% of which were currently residing in their country of birth. Age was collected categorically and, for analyses, bins were

assigned numbers from 1 through 6: 18–20 years (1), 21–24 years (2), 25–34 years (3), 35–44 years (4), 45–54 years (5), and 55+ years (6). Median age was 2.00 ($M$ = 2.20, $SD$ = 1.07, $Md$ = 1.00). Although some men and non-binary individuals did participate in the survey, only self-identified women are included in the current analyses.

## Respondents' use of sex–tech

**Meeting and connecting with sexual partners.** Study participants reported on whether they had ever used an app to find sexual partners and the types of partners sought. Options provided for types of sexual partners were: (1) "one night stands/hook-ups (one encounter/ date)", (2) "short-term relationships (a few meetings/dates)", (3) "a friends-with-benefits relationship (no romantic connection, but regular/frequent sex)", (4) "for chatting and sexting, but not to arrange a meeting/date/hook-up", (5) "a long-term relationship", (6) "Other", and (7) "I have not used an app to find a sexual partner"; more than one option could be selected. Participants reported whether they had ever sexted (defined as "sent someone content of a sexual nature") via text or an app. If yes, they reported the medium used: text, photos, audio/talking, or video; more than one option could be selected.

**Learning about sex and improving sexual relationships.** Participants reported whether they had ever used an app to learn more about sex or sexual intimacy. They also reported whether they had used an app or mobile device to improve their sexual relationship with a partner. If yes, they were presented with 12 options regarding their use. These options were: (1) "The app helped you explore new sexual experiences, like toys or positions", (2) "The app helped you stay connected when you could not see each other in person", (3) "The app helped you learn about ways to have safer sex", (4) "The app helped you feel more comfortable with your body or sexuality", (5) "The app helped you feel more comfortable with your partner's body," (6) "The app helped you learn what your partner finds arousing," (7) "The app helped your partner learn what you find arousing," (8) "The app helped you introduce other people into your sexual relationships," (9) "The app helped you learn about polyamory/open relationships/consensual non-monogamy," (10) "The app helped you feel more emotionally connected to your partner," (11) "The app did not help or wasn't useful," and (12) "The app did not help and it was detrimental to your sexual relationship." Participants could select multiple options.

**Tracking personal sexual health.** Participants reported whether they had ever used an app to track their sexual activity, "such as recording the days you have sex, whether you had an orgasm, or any other aspect of the experience." They also reported whether they had ever used an app to track sexual satisfaction or sexually transmitted infections.

## Gender Inequality Index

The Gender Inequality Index (GII; [18]) is a country-level measure of inequality developed by the United Nations Development Programme. The GII measures gender inequality by assessing three components within each country: (1) reproductive health measured by maternal mortality rates and adolescent birth rates; (2) women's empowerment, measured by the ratio of governmental seats held by women and the ratio of women and men aged 25 and older with some secondary education; and (3) economic status, measured by the proportion of labor force participation between men and women aged 15 and older. The GII is available for 159 countries and was developed to evaluate gender differences in essential areas of human development. The index ranges from 0 to 1, with higher values representing greater gender disparities. GII estimates were unavailable for 37 countries, comprising 4.5% ($n$ = 5,872 persons) of the study sample.

### Data analyses

All analyses were performed with SPSS, v. 26 [62]. The number of participants from each country represented in our sample is not proportional to that country's contribution to the total number of adult females globally. Therefore, to better reflect the global distribution of women and to facilitate regional and sub-regional comparisons, we weighted our data by country-specific adult female population size estimated by the World Bank [63]. Specifically, we calculated the percent contribution of each country to the world's adult female population (e.g., the United States accounted for 4.41% of the global total). We also calculated the percent of women that each country in the dataset contributes to the total sample size (e.g., the United States accounts for 31.51% of our sample of women). We then divided each country's percent of the world population of women (from World Bank data) by each country's percent of the women in our sample (e.g., for the United States, 4.41/31.51 = a weight of 0.14). We conducted all analyses—including frequency counts and regression analyses—using these weights. In addition to analyses of our total sample, we examined global variability by grouping countries into major regions and sub-regions based on the geoscheme employed by the United Nations [64].

To evaluate the possible impacts of gender inequality, we conducted 24 binary logistic regressions. The GII served as the predictor variable (continuous; mean-centered). Because technology use may be more prevalent in younger age groups, participant age was included as a control variable (mean-centered). For the outcome variables, we tested each of the five sought-after partner types, if they had never used an app to find a partner, whether they had sexted, whether they had used an app to learn about sex, whether they had used an app to improve their sexual relationships, each of the 12 response options for using mobile devices to improve a relationship, and whether they had used an app to track their sexual behavior, sexual satisfaction, or sexually transmitted diseases. We present all results (odds ratio) in terms of the behaviors of women in areas of higher gender inequality ('HGI women'; i.e., those living in countries with higher GII scores).

## Results

### Prevalence and reasons for use of mobile sex–tech

Results for the entire sample, and regional and sub-regional divisions, are presented in Table 1. Note that for Tables 1–3, colored "heat map" tables of these findings are included in S1–S3 Tables for easier interpretability.

### Meeting and connecting with sexual partners

About one-fifth (21.8%) of the global sample reported using mobile apps to find sexual partners (Table 2). This use varied by region: about one in five in Asia and Africa, one in four in Europe and the Americas, and one in three in Oceania. There is also marked within-region variability. Northern and Western Europe and North America resemble Oceania. Prevalence is less than 15% in North and West Africa and in the Caribbean.

Globally, the most commonly sought partners were short-term partners (9.0%), followed by chatting/sexting partners (8.7%), long-term partners (8.6%), and one-night stands/hook-ups (6.2%). Friends-with-benefits were the least sought partner (5.2%) in all regions except East Africa where friends-with-benefits were sought (8.1%) more than long-term partners (4.1%).

There are other marked regional differences. In all European regions, the most sought partner was a long-term partner (9.2%-18.1%); in North America, short- and long-term partners were equally sought (about 18%). In Oceania and Middle Africa, hook-ups are the most sought

**Table 1. Frequencies for All Sex-Tech Items for the Overall Sample.**

| Item | %N selecting yes |
|---|---|
| *Meeting and Connecting* | |
| Types of sexual partners found via app: | |
| One-night stands/hook-ups | 6.2% |
| Friends with benefits | 5.2% |
| Short-term partners | 9.0% |
| Chatting/sexting partners | 8.7% |
| Long-term partners | 8.6% |
| Have not used an app to find partners | 78.2% |
| Have you ever sexted? | 57.7% |
| Type of sext exchanged: | |
| Text | 52.6% |
| Photos | 43.0% |
| Audio/talking | 23.9% |
| Video | 22.9% |
| *Learning and Improving* | |
| Have you ever used an app to learn more about sex or sexual intimacy? | 19.4% |
| Have you used an app to improve your sexual relationship with a partner? | 10.9% |
| How did the app improve your sexual relationship: | |
| The app helped you explore new sexual experiences, like toys or positions | 3.6% |
| The app helped you stay connected when you could not see each other in person | 5.0% |
| The app helped you learn about ways to have safer sex | 1.9% |
| The app helped you feel more comfortable with your body or sexuality | 2.9% |
| The app helped you feel more comfortable with your partner's body | 2.0% |
| The app helped you learn what your partner finds arousing | 3.4% |
| The app helped your partner learn what you find arousing | 2.2% |
| The app helped you introduce other people into your sexual relationships | 0.4% |
| The app helped you learn about polyamory, open relationships, consensual non-monogamy | 0.4% |
| The app helped you feel more emotionally connected to your partner | 2.8% |
| The app did not help or wasn't useful | 0.6% |
| The app did not help and it was detrimental to your sexual relationship | 0.2% |
| *Tracking* | |
| Have you used an app to track your sexual activity? | 34.3% |
| Have you used an app to track sexual satisfaction? | 3.9% |
| Have you used an app to track sexually transmitted infections (STIs)? | 1.3% |

partners (about 18%). Chat/Sexting partners were the most sought in East Africa (16.2%), West Africa (6.6%) and South and West Asia (8.2% and 8.5%, respectively).

Over half (57.7%) of all respondents reported having engaged in sexting. This was consistent across all major regions (51.8%-68.6%). Across all regions, text (47.9%-61.6%) and photos (37.9%-56.7%) were the most common medium used for sexting.

## Learning about sex and improving sexual relationships

Nearly 20% of women reported having used an app to learn more about sex or sexual intimacy (Table 3). Regionally, the proportions were about 1 in 5 in Asia, the Americas and Europe, somewhat lower (17%) in Africa, and only 12.8% in Oceania. The highest rate was in Central Asia (29.8%), the lowest in East Africa (10.9%).

**Table 2. Frequencies of those selecting "yes" for all items related to meeting and connecting, by UN-designated regions and sub-regions.**

| | Sex–Tech Meeting and Connecting Items | | | | | | | | | | |
|---|---|---|---|---|---|---|---|---|---|---|---|
| | Types of partners found via apps | | | | | | | Type of sext exchanged | | | |
| | Ever used app to find sexual partner | One-night stands/hook-ups | Friends with Benefits | Chat/Sext partners | Short-term partners | Long-term partners | Ever sexted | Text | Photo | Audio | Video |
| **Regions and sub-regions** | | | | | | | | | | | |
| **Africa** | 19.1% | 5.1% | 4.1% | 9.7% | 7.0% | 4.8% | 65.3% | 58.1% | 45.5% | 26.2% | 22.6% |
| Eastern | 26.5% | 3.7% | 8.1% | 16.2% | 10.0% | 4.1% | 70.5% | 63.4% | 44.6% | 27.9% | 23.5% |
| Middle | 19.8% | 18.6% | 0% | 9.0% | 10.9% | 1.3% | 71.4% | 69.6% | 46.4% | 30.1% | 26.1% |
| Northern | 14.7% | 2.9% | 2.8% | 6.6% | 4.9% | 4.8% | 48.5% | 36.9% | 37.5% | 19.6% | 14.2% |
| Southern | 24.6% | 5.9% | 5.0% | 9.6% | 11.4% | 12.4% | 77.1% | 72.8% | 59.9% | 27.8% | 24.7% |
| Western | 12.6% | 1.8% | 2.1% | 4.7% | 2.6% | 5.9% | 65.2% | 58.1% | 48.7% | 26.6% | 24.9% |
| **Americas** | 28.3% | 8.8% | 7.5% | 9.8% | 13.2% | 12.6% | 68.6% | 61.6% | 56.7% | 25.4% | 29.2% |
| Caribbean | 14.8% | 4.2% | 1.6% | 3.3% | 5.8% | 9.2% | 66.9% | 58.1% | 53.4% | 25.9% | 21.9% |
| Central | 21.0% | 4.2% | 4.7% | 9.1% | 7.9% | 6.7% | 64.1% | 54.2% | 55.6% | 20.9% | 28.9% |
| South | 27.7% | 9.3% | 7.8% | 8.4% | 11.8% | 10.4% | 65.8% | 56.9% | 53.0% | 26.7% | 26.8% |
| Northern | 34.0% | 11.0% | 9.1% | 12.3% | 18.0% | 18.3% | 74.2% | 70.9% | 61.8% | 26.0% | 33.0% |
| **Asia** | 20.0% | 5.7% | 4.8% | 8.1% | 8.2% | 7.7% | 51.8% | 47.8% | 37.9% | 23.9% | 21.9% |
| Central | 19.1% | 1.4% | 0.7% | 0.5% | 0.9% | 5.0% | 18.1% | 15.6% | 12.3% | 4.7% | 4.0% |
| Eastern | 19.2% | 5.8% | 5.1% | 7.5% | 8.4% | 7.3% | 38.8% | 33.9% | 26.4% | 16.8% | 14.1% |
| Southern | 18.3% | 5.1% | 4.2% | 8.2% | 7.5% | 7.1% | 61.4% | 58.6% | 46.4% | 30.7% | 27.3% |
| South Eastern | 28.3% | 7.7% | 6.6% | 10.5% | 11.1% | 12.1% | 60.1% | 56.0% | 45.4% | 24.9% | 28.1% |
| Western | 17.3% | 5.0% | 3.8% | 8.5% | 7.0% | 5.4% | 52.1% | 46.8% | 36.2% | 22.4% | 21.0% |
| **Europe** | 27.4% | 7.3% | 5.8% | 9.2% | 10.9% | 14.0% | 63.2% | 58.0% | 48.3% | 16.9% | 19.4% |
| Eastern | 22.6% | 4.4% | 3.3% | 6.1% | 8.0% | 13.4% | 63.0% | 56.0% | 49.1% | 14.6% | 16.8% |
| Northern | 37.8% | 11.4% | 7.8% | 14.4% | 17.7% | 18.1% | 72.5% | 68.3% | 57.0% | 21.2% | 27.8% |
| Southern | 23.1% | 5.9% | 5.5% | 8.4% | 8.2% | 9.2% | 61.9% | 56.6% | 46.7% | 18.8% | 20.4% |
| Western | 32.8% | 10.7% | 8.7% | 11.8% | 13.8% | 16.4% | 59.5% | 56.6% | 43.8% | 16.8% | 18.4% |
| **Oceania[a]** | 36.6% | 17.9% | 9.9% | 12.2% | 15.9% | 13.4% | 61.7% | 58.3% | 50.3% | 23.1% | 22.7% |

[a]The United Nations specifies four sub-regions of Oceania: Australia and New Zealand, Melanesia, Micronesia, Polynesia. In the current sample, only 17 participants lived in the latter three sub-regions. Due to the small sample size, we report only on the major region of Oceania rather than its sub-regions.

Globally, 11% reported having used an app or mobile device to improve their sexual relationship. There was only slight variation in this proportion across the five major regions (10% to 15%) nor within Europe. However, across the subregions within Africa, the Americas, and Asia, the range of sex-tech use for improving a sexual relationship was two-fold and more.

Of those who had used an app to improve their sexual relationship, the three most common reasons were staying connected with their sexual partner when they could not see each other in person (5.0%); facilitating exploration of new sexual experiences, such as new sex toys or positions (3.6%); and helping them to learn what their partner finds arousing (3.4%). The least common reasons for use were to introduce other people into the sexual relationship (0.4%) and to learn about polyamory, open relationships, and consensual non-monogamy (0.4%). Very few women reported that the app they used to try to improve their sexual relationships was detrimental (0.2%) or not useful (0.6%).

In nearly all regions and sub-regions, staying connected when apart was the most commonly selected reason for app use with a sexual partner. However, in Middle Africa, the most commonly selected reason (10.9%) was helping the user to feel more comfortable with their

**Table 3. Frequencies of those selecting "yes" to all items relating to learning and improving, by UN-designated regions and sub-regions.**

| | Sex–Tech Learning and Improving Items | | | | | | |
|---|---|---|---|---|---|---|---|
| | Used an app to. . . | | | The app improved sexual relationship by helping to. . . | | | |
| | Learn about sex | Improve sexual relationship | Explore new sexual experiences | Stay connected when apart | Learn about safer sex | Feel comfortable with own body/ sexuality | Feel comfortable with partner's body |
| **Regions and sub-regions** | | | | | | | |
| **Africa** | 17.1% | 13.6% | 5.1% | 4.3% | 2.4% | 3.7% | 3.3% |
| Eastern | 10.9% | 12.2% | 4.4% | 4.8% | 2.3% | 2.8% | 1.7% |
| Middle | 21.7% | 16.1% | 0.9% | 1.9% | 0.6% | 1.5% | 10.9% |
| Northern | 13.0% | 7.8% | 3.1% | 4.6% | 1.4% | 3.4% | 1.6% |
| Southern | 19.1% | 13.3% | 3.6% | 5.9% | 2.8% | 4.0% | 2.2% |
| Western | 24.1% | 17.9% | 9.4% | 4.4% | 3.7% | 5.6% | 3.0% |
| **Americas** | 19.9% | 10.9% | 3.8% | 5.6% | 1.4% | 3.5% | 1.7% |
| Caribbean | 18.8% | 7.6% | 2.2% | 4.9% | 1.0% | 2.2% | 0.9% |
| Central | 22.4% | 8.1% | 2.9% | 3.5% | 1.6% | 2.5% | 1.2% |
| South | 19.9% | 9.0% | 3.9% | 3.7% | 1.4% | 2.8% | 1.5% |
| Northern | 18.8% | 14.7% | 4.4% | 9.0% | 1.3% | 5.0% | 2.3% |
| **Asia** | 20.1% | 10.3% | 3.1% | 5.1% | 2.0% | 2.6% | 1.8% |
| Central | 29.8% | 24.6% | 0.5% | 0.5% | 0.0% | 0.5% | 0.2% |
| Eastern | 28.1% | 6.7% | 3.2% | 2.5% | 2.3% | 2.4% | 1.4% |
| Southern | 14.3% | 12.2% | 2.6% | 7.0% | 1.5% | 2.7% | 2.2% |
| South Eastern | 17.0% | 13.0% | 4.2% | 7.4% | 3.3% | 3.4% | 1.8% |
| Western | 15.9% | 8.0% | 3.2% | 3.9% | 1.6% | 2.5% | 1.4% |
| **Europe** | 19.5% | 10.0% | 3.5% | 5.0% | 0.7% | 2.6% | 1.2% |
| Eastern | 23.4% | 10.0% | 3.5% | 4.5% | 0.8% | 2.6% | 1.4% |
| Northern | 14.0% | 11.0% | 3.4% | 6.5% | 0.8% | 3.1% | 1.3% |
| Southern | 17.8% | 7.8% | 3.1% | 3.4% | 0.7% | 1.9% | 0.9% |
| Western | 17.8% | 11.4% | 3.9% | 6.0% | 0.4% | 2.6% | 1.1% |
| **Oceania** | 12.8% | 15.2% | 8.7% | 5.7% | 5.7% | 8.3% | 6.3% |
| | Sex–Tech Learning and Improving Items | | | | | | |
| | The app improved sexual relationship by helping to. . . | | | | | | |
| | Learn what arouses partner | Partner learn what arouses me | Introduce others into sexual relationship | Learn about polyamory, open relationships, non-monogamy | Feel more emotionally connected | The app did not help or wasn't useful | The app was detrimental |
| **Regions and sub-regions** | | | | | | | |
| **Africa** | 3.6% | 1.2% | 0.1% | 0.1% | 2.7% | 0.9% | 0.1% |
| Eastern | 2.9% | 0.8% | 0.1% | 0.2% | 3.9% | 0.2% | 0.4% |
| Middle | 2.6% | 1.3% | 0.0% | 0.6% | 1.9% | 0.6% | 0.0% |
| Northern | 2.3% | 0.7% | 0.3% | 0.2% | 1.1% | 1.2% | 0.0% |
| Southern | 4.1% | 2.8% | 0.3% | 0.0% | 3.4% | 1.1% | 0.2% |
| Western | 5.5% | 1.8% | 0.0% | 0.2% | 2.6% | 1.6% | 0.0% |
| **Americas** | 3.6% | 2.5% | 0.4% | 0.3% | 2.7% | 0.5% | 0.1% |
| Caribbean | 2.4% | 1.4% | 0.1% | 0.1% | 1.3% | 0.3% | 0.1% |
| Central | 2.6% | 1.7% | 0.2% | 0.2% | 1.7% | 0.7% | 0.2% |
| South | 3.1% | 1.9% | 0.4% | 0.2% | 1.9% | 0.5% | 0.1% |
| Northern | 4.7% | 3.8% | 0.4% | 0.5% | 4.3% | 0.6% | 0.2% |
| **Asia** | 3.3% | 2.3% | 0.5% | 0.5% | 2.9% | 0.5% | 0.2% |

*(Continued)*

**Table 3.** (Continued)

| | | | | | | | |
|---|---|---|---|---|---|---|---|
| Central | 23.3% | 0.2% | 0.0% | 0.0% | 0.2% | 0.7% | 0.0% |
| Eastern | 1.9% | 1.6% | 0.7% | 0.7% | 1.6% | 0.2% | 0.0% |
| Southern | 3.7% | 2.7% | 0.4% | 0.4% | 4.0% | 0.8% | 0.4% |
| South Eastern | 3.4% | 2.9% | 0.4% | 0.5% | 3.2% | 0.8% | 0.0% |
| Western | 2.9% | 2.3% | 0.1% | 0.3% | 2.2% | 0.2% | 0.1% |
| **Europe** | 3.1% | 2.2% | 0.2% | 0.3% | 2.6% | 0.6% | 0.2% |
| Eastern | 3.1% | 2.1% | 0.2% | 0.2% | 2.7% | 0.6% | 0.1% |
| Northern | 3.2% | 2.7% | 0.3% | 0.3% | 2.8% | 0.5% | 0.1% |
| Southern | 2.5% | 1.7% | 0.3% | 0.3% | 1.8% | 0.4% | 0.1% |
| Western | 3.5% | 2.6% | 0.3% | 0.3% | 3.0% | 0.8% | 0.3% |
| **Oceania** | 7.8% | 7.6% | 5.6% | 0.3% | 7.7% | 0.6% | 0.1% |

partner's body (an option that ranked quite low in all other locales). Helping the user to learn what arouses their partner was the top selection in Central Asia (23.3%); all other options were 0.5% or less; this response pattern was not found in any other subregion. Only respondents in Oceania indicated appreciable app use for introducing others into their existing sexual relationship (5.6%).

## Tracking personal sexual health

About a third of the respondents (34.3%) reported that they had used an app to track their own sexual activity (Table 4). Much smaller proportions reported that they had used an app to track sexual satisfaction (3.9%) or sexually transmitted infections (1.3%). Across major regions, differences in prevalence were modest: sexual activity (31.3%-39.9%), sexual satisfaction (3.1%-7.1%), and STIs (0.7%-1.9%). In the Americas and Europe there was modest variability across subregions. In contrast, 13.2% of participants in North Africa had tracked their sexual activity compared to ~30%-50% in other African sub-regions. East Asia respondents were nearly twice as likely as West Asia respondents to record sexual activity (37.9% vs. 20.1%).

## Gender inequality and mobile sex–tech use

We present the results in terms of the behaviors of women in areas of higher gender *in*equality ('HGI women'; i.e., those living in countries with higher GII scores). Odds ratios for all tests are presented in Table 5. Note that because GII is a continuous variable, significant (at *p* <0.05) odds ratios are interpreted here as: with every one-point increase in GII (with increasing *in*equality), women are [odds ratio] more/less likely to engage in the behavior.

　HGI women were less likely to have used mobile apps to find a sexual partner, with each GII one-point increase corresponding with a 40% decrease in likelihood of having done so. They were specifically less likely to have used a mobile app to find one-night-stands/hook-up-partners, friends with benefits, short- and long-term partners, such that with each one-point increase in GII, the likelihood of searching for these partners decreased by 12%, 18%, 24%, and 37% respectively. However, there was no difference in using an app to find someone to chat/sext with (i.e., although HGI women were less likely to use mobile apps to find a partner, there was no difference in the reported prevalence of using apps to find a chat/sext partner). In regard to sexting, HGI women were nearly four times *more* likely than women in areas of higher equality to report having engaged in sending and receiving sexts, such that with one-point GII increases, likelihood of sexting also increased nearly four-fold.

**Table 4. Frequencies of those selecting "yes" to all items relating to tracking by UN-designated regions and sub-regions.**

| | Sex–Tech Tracking Items | | |
|---|---|---|---|
| | Have you used an app to track... | | |
| | Sexual activity | Sexual satisfaction | Sexually transmitted infections |
| *Regions and sub-regions* | | | |
| **Africa** | 31.3% | 6.5% | 1.3% |
| Eastern | 28.6% | 8.1% | 2.5% |
| Middle | 49.7% | 3.8% | 0.0% |
| Northern | 13.2% | 1.9% | 1.0% |
| Southern | 35.1% | 2.8% | 0.8% |
| Western | 36.7% | 9.5% | 0.7% |
| **Americas** | 39.9% | 4.2% | 1.9% |
| Caribbean | 33.6% | 5.9% | 5.3% |
| Central | 40.1% | 4.4% | 2.0% |
| South | 35.5% | 4.7% | 2.3% |
| Northern | 45.6% | 3.2% | 1.0% |
| **Asia** | 32.9% | 3.1% | 1.2% |
| Central | 36.6% | 0.2% | 0.0% |
| Eastern | 37.9% | 3.8% | 0.8% |
| Southern | 28.8% | 1.8% | 1.6% |
| South Eastern | 36.5% | 4.2% | 1.4% |
| Western | 20.1% | 6.3% | 0.6% |
| **Europe** | 38.5% | 3.5% | 1.1% |
| Eastern | 39.0% | 4.2% | 1.2% |
| Northern | 35.0% | 1.9% | 0.9% |
| Southern | 37.4% | 4.1% | 1.1% |
| Western | 40.3% | 3.0% | 1.2% |
| **Oceania** | 39.9% | 7.1% | 0.7% |

HGI women were about half as likely to report that they had used mobile technology to learn about sex, but were nearly 1.5 times *more* likely to have used an app to improve their sexual relationships. Looking at differences in specific reasons for using apps to improve relationships, most (8 of 12) comparisons were non-significant. HGI women were more likely than women in areas of lower inequality to report having used mobile technology for the following reasons: (1) helped you to stay connected when you could not see each other in person, (2) helped you feel more comfortable with your body or sexuality, (3) helped you learn what your partner finds arousing, and (4) helped you feel more emotionally connected to your partner. With each GII one-point increase, the likelihood of selecting these reasons increased by 18%, 17%, 7%, and 13% respectively. Last, HGI women were around half as likely to report that they have tracked their sexual activity, but were slightly (~10%) more likely to report having tracked their sexual satisfaction. There was no difference in tracking STIs.

## Discussion

Sex-related mobile tools are rapidly becoming ubiquitous, and the mobile aspect makes these tools accessible to a diversity of populations: places with and without access to evidence-based sex education or healthcare, with and without freedom of sexual expression, and across a spectrum of dating and partnership norms. Our understanding of the impacts of these tools is in

**Table 5. Odds ratios and 95% confidence intervals for GII on each sex–tech item, controlling for participant age.**

| | Predictor Variables | |
|---|---|---|
| | **Age** | **GII** |
| **Item** | OR [95% CI] | OR [95% CI] |
| *Meeting and Connecting* | | |
| Have you used an app to find a sexual partner? | 1.08 [1.07, 1.10]*** | 0.60 [0.57, 0.64]*** |
| Types of sexual partners found via app: | | |
| One-night stands/hook-ups | 1.04 [1.03, 1.06]*** | 0.88 [0.83, 0.94]*** |
| Friends with benefits | 1.05 [1.03, 1.06]*** | 0.82 [0.77, 0.87]*** |
| Short-term partners | 1.10 [1.08, 1.01]*** | 0.76 [0.71, 0.81]*** |
| Chatting/sexting partners | 0.98 [0.96, 0.98]*** | 0.95 [0.90, 1.02] |
| Long-term partners | 1.12 [1.11, 1.14]*** | 0.63 [0.60, 0.67]*** |
| Have you sent or received a sext? | 0.92 [0.91, 0.93]*** | 3.78 [3.54, 4.02]*** |
| *Learning and Improving* | | |
| Have you used an app to learn about sex? | 0.95 [0.94, 0.96]*** | 0.43 [0.40, 0.46]*** |
| Have you used an app to improve your sexual relationship? | 1.02 [1.01, 1.04]*** | 1.41 [1.33, 1.50]*** |
| How did the app improve your sexual relationship: | | |
| The app helped you explore new sexual experiences, like toys or positions | 1.04 [0.97, 0.99]*** | 1.03 [0.97, 1.01] |
| The app helped you stay connected when you could not see each other in person | 1.01 [0.99, 1.02] | 1.18 [1.11, 1.26]*** |
| The app helped you learn about ways to have safer sex | 1.01 [0.99, 1.02] | 1.01 [0.95, 1.08] |
| The app helped you feel more comfortable with your body or sexuality | 1.01 [1.00, 1.02] | 1.00 [0.94, 1.07] |
| The app helped you feel more comfortable with your partner's body | 0.99 [0.98, 1.01] | 1.17 [1.10, 1.25]*** |
| The app helped you learn what your partner finds arousing | 1.00 [0.99, 1.02] | 1.07 [1.01, 1.14]* |
| The app helped your partner learn what you find arousing | 1.01 [0.99, 1.02] | 1.02 [0.96, 1.09] |
| The app helped you introduce other people into your sexual relationships | 1.01 [1.00, 1.02] | 0.99 [0.93, 1.05] |
| The app helped you learn about polyamory, open relationships, consensual non-monogamy | 1.01 [0.99, 1.02] | 0.97 [0.92, 1.04] |
| The app helped you feel more emotionally connected to your partner | 1.01 [1.00, 1.03]* | 1.13 [1.06, 1.20]*** |
| The app did not help or wasn't useful | 1.02 [1.01, 1.03]† | 1.03 [0.97, 1.10] |
| The app did not help and it was detrimental to your sexual relationship | 1.01 [1.00, 1.02] | 1.02 [0.96, 1.09] |
| *Tracking* | | |
| Have you used an app to track your sexual activity? | 1.19 [1.18, 1.20]*** | 0.49 [0.46, 0.52]*** |
| Have you used an app to track sexual satisfaction? | 1.00 [0.99, 1.01] | 1.10 [1.03, 1.17]** |
| Have you used an app to track sexually transmitted infections (STIs)? | 1.01 [1.00, 1.02] | 1.02 [0.96, 1.09] |

***$p < .001$
**$p < .01$
*$p < .05$
†$p = .001$

its infancy. To our knowledge, and although not representative of women worldwide, this study provides the most comprehensive global data on sex–tech use thus far, and is the first to examine women's engagement in sex-related mobile technology in locations with greater gender disparities.

Globally, in our sample the most common uses for mobile sex–tech, in descending order, were sexting, tracking sexual activity, finding sexual partners, and learning about sex. Respondents in countries with greater gender inequality (i.e., a higher GII score) were more likely to have sexted, to have engaged with mobile apps/technology to try and improve their sexual relationships, and to have tracked their sexual satisfaction, but were less likely to participate in most other sex–tech engagement measured here.

## Finding partners and sexting

Online dating appears to be a successful medium for pursuing enduring romantic relationships as well as casual sex relationships [27,40,65–69]. In studies assessing motives behind online dating, the majority of respondents report wanting to find a romantic relationship, with smaller numbers specifically seeking casual sex relationships [70,71]. Although there are some financial and physical-safety risks associated with in-person meetings subsequent to an online match (e.g., [27,32,70]), 66% of online daters in the U.S. have gone on a date with someone met through a website or app [28]. Of these, 23% met their spouse or long-term relationship partner via these services. Other research conducted with Belgian Tinder users reported that roughly one-third of offline encounters led to casual sex, and over one-quarter led to long-term romantic partnerships [30].

Globally, about one-fifth (21.8%) of our sample reported using apps to find sexual partners, and were most commonly seeking short-term partners, followed closely by long-term partners and partners strictly for chatting/sexting. There are, however, marked regional differences. Because there have been only a few prior studies, only a few comparisons are possible. For example, based on data collected in 2016, over one in five adults aged 18 to 44 years in the United States had tried online dating (including app and web; [29]). Our data, showing use for app-based methods alone, suggest about one in three North American respondents have tried on-line dating, a finding in line with the increasing trend in the use of online dating apps (vs. web) and reflective of diminishing online dating taboos [27].

Women in countries with greater gender inequality were less likely to have ever used an app to find a sexual partner. These data likely reflect gender differences in relationship autonomy or decision-making power, as apps require active public participation in partner-choosing from both genders. Other factors may include perceived safety (regions of less gender inequality also have higher rates of sexual violence; [72]), differences in community dating norms and traditions, or cultural stigma toward online dating. While the majority of North Americans have come to view online dating as a primarily positive tool, its users were once perceived as socially incompetent, desperate, immature, and self-centered [25–28]. Because these technologies are newer outside North America, stigma towards online dating may be quite high. However, it is important to note that on average—with sample weighting—one in five women were using online dating apps to find some type of sexual partner. Though the reports were lower with greater gender inequality, these results still provide evidence that mobile sex–tech is spreading beyond North America and beyond our common sampling frame in the related literature.

Seeking partners specifically for chatting or sexting was the second most common partnership facilitated by online dating services, and no significant difference was found in regards to gender inequality. It may be that finding partners for this type of interaction is a common 'first step' in exploring online dating, as it allows for interacting without any in-person interaction that could have various types of risk. Further, over half of the women in our sample had engaged in sexting, and surprisingly, women in areas of greater inequality were nearly four times more likely than women in areas of more equality to report having done so. Sexting was most common in the Americas (~67%), and least common, but still very prevalent, in Asia (~52%). Our findings show a higher rate of sexting behavior than previous research done in the U.S.—a 2016 survey found only one in five adults had reported having engaged in sexting [37]. This may be due to sampling from a more sex–tech-savvy pool, as well as an increasing commonality of sexting, with the increase of technological integration and accessibility of apps providing anonymity (e.g., SnapChat).

### Learning about sex and improving sexual relationships

Increased education for girls and women has been associated with lower fertility (e.g., [73]), more contraceptive use [74], and later ages at first sex, first marriage, and first birth [75] in developing countries. With previous research showing that lack of autonomy negatively impacts women's access to health education and health services [76], the use of mobile devices to learn about sex seemed a possibly beneficial avenue for women to educate themselves, considering the potential for relative privacy in what one searches for or does on a device. While nearly one in five women overall reported having used an mobile device or app to learn more about sex or sexual intimacy, women living with greater gender inequality were less likely to report doing so, but were actually more likely to report using this type of technology to *improve* their sexual relationship. Taken together, these findings suggest that—because gender inequality is a significant barrier to women's ability to control their own sexual and reproductive lives [77], and because areas that are more male-dominated tend to view women's sexuality as belonging to her husband or partner—women are likely learning the specifics of sex via their partner and the partner's methods or desires. Having this foundational knowledge already, these women are more interested in improving their sexual lives and relationships. They were just as likely as women in places of more equality to use mobile devices/apps to learn about safer sex, to explore new sexual experiences like new positions or sexual aids, to help them feel more comfortable with their own body, and to help their partners learn what they (the women) find arousing. Meanwhile, these women were even more likely to report having used an app to help them stay connected to their sexual partner when they could not see each other in person, to feel more emotionally connected to their partner, to feel more comfortable with their partner's body, and to learn what arouses their partner. This difference in women's desire to 'learn' versus 'improve' may be especially useful to future researchers who are working to provide sex education to women in areas like this, potentially increasing success of the intervention by creating a relationship improvement tool that incorporates elements of emotional connectedness in addition to sexuality, rather than focusing on straightforward sex education.

### Tracking personal sexual health

Sexual activity was by far the most commonly tracked item, with 34% of women reporting having tracked their behaviors. Tracking sexual satisfaction (3.9%) and sexually transmitted infections (1.3%) was much less prevalent. However, reports of tracking sexual activity and satisfaction were impacted by the GII, such that women in countries with greater gender inequality were less likely to report tracking their activity than were women in more equal countries, but were more likely to report having tracked sexual satisfaction. The latter finding may be related to the desire to improve one's sexual relationship, using markers of satisfaction to monitor one's progress in improvement. However, lower prevalence of tracking sexual activity could provide information for an appropriate starting point for future interventionists: introducing easy-to-use tools that help to track sexual activity and providing education on how such tracking can prevent or facilitate the resulting outcomes.

### Strengths and limitations

The current study is the largest known survey of women's sex–tech engagement, and the first to explore this topic on a global level. However, it should be noted that our survey respondents were not a representative sample, and were most likely to have participated in the study after signing up to the newsletter of a sex-positive women's health app or via the social media accounts of a well-known sex research institute. People who used that particular app or who

followed those social media accounts may have had a pre-established interest in sexual health topics greater than that of the general population. Our participants may also have been more self-aware of their sexual health than a more representative sample, although awareness does not necessarily indicate differences in behavior. Anthropological studies at the community level are needed to answer these questions. Thus, our findings provide a foundation for more detailed exploration in specific contexts.

There were some additional limitations that may have contributed to sampling bias. Although the Clue app is available in 15 languages, respondents in countries with first languages not covered in that list would be using an app in a second language, such as English, reflecting a higher educational background. Further, with the exception of Brazil, countries ranking highly on the GII scale (low gender equality) also tended to have far fewer respondents, which may have skewed the current results. Countries with higher GII are likely to be more conservative, especially with regards to sexuality. This likely resulted in a selection bias, skewing our sample toward more liberal sexual attitudes and behaviors.

In terms of measurement, respondents from countries that are more conservative or less Westernized may have been heretofore unfamiliar with specific behaviors or relationship types, such as friends with benefits or one-night stands. Although we cannot know if this is true, or in what specific countries this would be the case, this limitation may have inadvertently led to inaccurate data. Similarly, certain sex–tech behaviors may be more strongly influenced by factors other than, but relating to, the GII such as socioeconomic status, religion and religiosity, interaction with the Western world and Western media, and population density, among many others that were not measured here. Future research would benefit from the inclusion of these variables and other individual-difference factors that could impact women's abilities to own a smartphone and have private access to the contents of their smartphone, women's knowledge of apps with these capabilities, and women's risk of harm or punishment from engaging in these behaviors.

## Future directions

Although our data have generated a wealth of knowledge for the first foray into global sex–tech use by women, the impact that these sex–tech tools have in challenging and compensating for gender inequality is still unclear. As such, there are several avenues for future research. First, because we found such variance by GII, research on global sex–tech tools and how they may be challenging taboos and gender roles (dating and gender power balance, autonomy, partner choice, etc.) is needed. Similarly, while we know that internet-based materials can improve adolescents' understanding of sexual health knowledge, we do not know how information gleaned from the internet or tracking one's sexual health behavior and symptomology improves health outcomes (sexual or otherwise) or satisfaction with one's sexual life. We also do not know the ratio of accurate versus inaccurate sexual health information accessed online, and whether internet users attempt to verify the information. Thus, research is needed to determine health benefits across GII and across healthcare models [78], and the impact of incorrect sexual health information on women's sexual lives.

Second, we have hypothesized throughout this manuscript that many of our results could indicate that respondents view sex–tech or internet-based mediums as safe spaces in which to explore and express their sexuality, because these venues offer anonymity. Such anonymity would be valuable in cultures with substantial sexual taboos or punishment for women who engage in sexual behaviors. We have also assumed that areas of greater gender inequality offer no or inadequate sexual education, prompting women to turn to mobile technologies for information. We know of no data exploring attitudes toward sex–tech or global

understandings of motives behind women's use of mobile technologies to gain sexual health information, nor do we know of any data on sexual education around the world. Future research would benefit from incorporating these investigations into their studies of women and sex–tech use.

## Conclusion

Women around the world reported engaging with sex–tech to find partners of varying degrees of sexual expectations and commitment, to communicate with their partners, to learn about sex and intimacy, to improve their sexual relationships, and to track facets of their sexual experiences. These results indicate that mobile sex–tech is becoming a global phenomenon, albeit in its early stages in developing areas. As such, better understandings of women's motivations for engaging with sex–tech, the quality and accuracy of online and app-based sexual information, and how such engagement has been beneficial in different global contexts is of great importance. Women's interactions with app-based instruction, tracking, diagnostic support, and accessible sexual health information may be of particular importance to women's health professionals, sex educators, and interventionists with interest in improving the lives of women around the world via innovative means. Our findings provide the foundation for such future research and application, demonstrating that women worldwide are turning to sex–tech as a means of connecting, learning, reflecting, and improving their sexual lives.

## Supporting information

**S1 Data. Data file including all variables examined in the current paper.**
(SAV)

**S1 Table. Table 1 modified with "heat map" coloring.** Heat map coloring provided for ease of interpretability. Ranks of responses within a sample (i.e., across each row) are color coded: high% to low% are red, orange, yellow, green, blue.
(XLSX)

**S2 Table. Table 2 modified with "heat map" coloring.** Heat map coloring provided for ease of interpretability. Ranks of responses within a sample (i.e., across each row) are color coded: high% to low% are red, orange, yellow, green, blue.
(XLSX)

**S3 Table. Table 3 modified with "heat map" coloring.** Within regions, red = highest percent responding YES to tracking that facet of sexual health.
(XLSX)

**S1 Survey Tool. Full survey instrument.**
(DOCX)

## Author Contributions

**Conceptualization:** Amanda N. Gesselman, Anna Druet, Virginia J. Vitzthum.

**Data curation:** Amanda N. Gesselman, Anna Druet, Virginia J. Vitzthum.

**Formal analysis:** Amanda N. Gesselman.

**Investigation:** Amanda N. Gesselman, Anna Druet, Virginia J. Vitzthum.

**Methodology:** Amanda N. Gesselman, Anna Druet, Virginia J. Vitzthum.

**Project administration:** Amanda N. Gesselman, Anna Druet, Virginia J. Vitzthum.

**Software:** Amanda N. Gesselman.

**Supervision:** Anna Druet, Virginia J. Vitzthum.

**Validation:** Anna Druet.

**Visualization:** Amanda N. Gesselman, Anna Druet, Virginia J. Vitzthum.

**Writing – original draft:** Amanda N. Gesselman.

**Writing – review & editing:** Anna Druet, Virginia J. Vitzthum.

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
