## [Decision Letter · Decision Letter 0]

27 May 2020

PONE-D-20-06636

Sex at their fingertips: 

A global analysis of gender-inequality and the use of sex–tech apps

PLOS ONE

Dear Dr. Gesselman,

Thank you for submitting your manuscript to PLOS ONE. After careful consideration, we feel that it has merit but does not fully meet PLOS ONE’s publication criteria as it currently stands. Therefore, we invite you to submit a revised version of the manuscript that addresses the points raised during the review process.

The two reviewers had very different responses to your article submission. In your revision, please respond specifically to the methodological critiques given, including responding to each point of the critical review and how you accounted for potential bias in your subject selection.

Thank you.

We look forward to receiving your revised manuscript.

Kind regards,

Amy Michelle DeBaets, PhD

Academic Editor

PLOS ONE

Journal Requirements:

2. Please include additional information regarding the survey or questionnaire used in the study and ensure that you have provided sufficient details that others could replicate the analyses.

For instance, if you developed a questionnaire as part of this study and it is not under a copyright more restrictive than CC-BY, please include a copy, in both the original language and English, as Supporting Information.

3. Please modify the title to ensure that it is meeting PLOS’ guidelines (https://journals.plos.org/plosone/s/submission-guidelines#loc-title).

In particular, the title should be "specific, descriptive, concise, and comprehensible to readers outside the field" and in this case it is not informative and specific about your study's scope and methodology.

Please ensure you amend both the title on the online submission form (via Edit Submission) and the title in the manuscript so that they are identical.

4. Thank you for providing the following Funding Statement: 

'Yes. The funders, the company BioWink—where AD was employed but uninvolved—collected/paid to collect the data. They were not involved in design, analysis, or preparation of the manuscript.'

We note that one or more of the authors is affiliated with the funding organization, indicating the funder may have had some role in the design, data collection, analysis or preparation of your manuscript for publication; in other words, the funder played an indirect role through the participation of the co-authors.

a. If the funding organization did not play a role in the study design, data collection and analysis, decision to publish, or preparation of the manuscript and only provided financial support in the form of authors' salaries and/or research materials, please review your statements relating to the author contributions, and ensure you have specifically and accurately indicated the role(s) that these authors had in your study in the Author Contributions section of the online submission form. Please make any necessary amendments directly within this section of the online submission form.  Please also update your Funding Statement to include the following statement: “The funder provided support in the form of salaries for authors [insert relevant initials], but did not have any additional role in the study design, data collection and analysis, decision to publish, or preparation of the manuscript. The specific roles of these authors are articulated in the ‘author contributions’ section.”

If the funding organization did have an additional role, please state and explain that role within your Funding Statement.

5. Please include captions for your Supporting Information files at the end of your manuscript, and update any in-text citations to match accordingly. Please see our Supporting Information guidelines for more information: http://journals.plos.org/plosone/s/supporting-information

Reviewers' comments:

Reviewer's Responses to Questions

**Comments to the Author**

1. Is the manuscript technically sound, and do the data support the conclusions?

Reviewer #1: Partly

Reviewer #2: Yes

2. Has the statistical analysis been performed appropriately and rigorously? 

Reviewer #1: Yes

Reviewer #2: Yes

3. Have the authors made all data underlying the findings in their manuscript fully available?

Reviewer #1: No

Reviewer #2: Yes

4. Is the manuscript presented in an intelligible fashion and written in standard English?

Reviewer #1: Yes

Reviewer #2: Yes

5. Review Comments to the Author

Reviewer #1: Dear authors,

I congratulate you on this very interesting work. While the general topic of the research is clear, I believe that there is a significant risk of bias in this research, which ultimately raises some doubts on its validity.

The presence of Higher gender inequality (HGI) is generally associated with reduced availability of devices for dating apps and reduced possibilities for "open" relationships. Research on the topic of religion has indeed proven that "strict" beliefs are more often associated with the consideration of sex as a "dirty" act, therefore reducing pre-marital sex; likewise, chastity before marriage is still considered a "virtue" in many societies and even in parts of the Western world, such as some European countries, this is a recurring theme.

The risk of selection bias has to be considered in this research. The methods for disseminating the questionnaire - i.e. "Clue’s newsletter, website, and social media accounts, and the social media accounts of the Kinsey Institute" - likely limit the possibility of a representative sample; similarly, it is likely that those who actually took the time to fill the questionnaire were more self-aware of their sexual health.

I believe all these items should be explored more carefully.

Reviewer #2: This is a very fine paper and I don't think any major revisions are necessary. The topic is fairly novel and interesting, uses methods that are fairly standard in the field, and is an easy read. Kudos to the authors for a well done manuscript.

6. PLOS authors have the option to publish the peer review history of their article (what does this mean?). If published, this will include your full peer review and any attached files.

Reviewer #1: No

Reviewer #2: No

---

## [Author Response · Author response to Decision Letter 0]

3 Aug 2020

Please see our response to reviewer letter in the uploaded attachments for better formatting.

Reviewer #1

Reviewer #1: Dear authors,

I congratulate you on this very interesting work. While the general topic of the research is clear, I believe that there is a significant risk of bias in this research, which ultimately raises some doubts on its validity.

The presence of Higher gender inequality (HGI) is generally associated with reduced availability of devices for dating apps and reduced possibilities for "open" relationships. Research on the topic of religion has indeed proven that "strict" beliefs are more often associated with the consideration of sex as a "dirty" act, therefore reducing pre-marital sex; likewise, chastity before marriage is still considered a "virtue" in many societies and even in parts of the Western world, such as some European countries, this is a recurring theme. The risk of selection bias has to be considered in this research. The methods for disseminating the questionnaire - i.e. "Clue’s newsletter, website, and social media accounts, and the social media accounts of the Kinsey Institute" - likely limit the possibility of a representative sample; similarly, it is likely that those who actually took the time to fill the questionnaire were more self-aware of their sexual health.

I believe all these items should be explored more carefully.

Thank you to Reviewer 1 for their time and effort in reviewing our paper and providing this important feedback. We have worked to address these points more clearly. 

Your overall point is that there is selection bias in our sampling, and we agree. We’ve added statements about our sample being non-representative throughout, in both the Introduction, Methods, and Discussion. For example, we added ‘a non-representative sample’ to this sentence where we introduce the current study on pg. 4:

“In a non-representative sample of 130,885 women from 191 countries, we asked about their personal use of mobile sex–tech for meeting and connecting with sexual partners, for learning about sex and improving sexual relationships, and for tracking personal sexual health.”

And again in the first paragraph of the Discussion on pg. 23:

“To our knowledge, and although not representative of women worldwide, this study provides the most comprehensive global data on sex–tech use thus far, and is the first to examine women’s engagement in sex-related mobile technology in locations with greater gender disparities.” 

Regarding your first point about locations with higher gender inequality likely having reduced access to smartphone devices, we certainly agree that smartphone use is less saturated in countries with lower income. However, there are “rich” countries with high gender inequality, as in Saudi Arabia, so we wish to be cautious in generalizing here. Additionally, smartphone access has been spreading to lower and middle income countries over the last several years, with lower-priced models available in these countries that fit the price points of the local economy (see “mobile-first markets”). We’ve added more detailed information about the presence of smartphones in emerging economies to help clarify how widespread such use is in non-Westernized countries on pg. 4:

“Additionally, smartphone access has rapidly spread around the world. A 2018 study conducted by the Pew Research Center showed that 76% of people in advanced economies (e.g., United States, United Kingdom, Australia, Israel, South Korea) and 45% of people in emerging economies (e.g., India, Indonesia, Kenya, Nigeria, Tunisia) have smartphones, although this tends to be skewed toward more ownership in younger populations [13]. With rising access to smartphones, and a near-universal motivation to seek romantic and sexual connection, mobile technologies that serve to enhance or advance these relationships are likely to have spread beyond those few regions that have been studied to date.”

Regarding your second point about selection bias around social norms in conservative cultures, we again agree that these norms, such as stricter gender roles, may shape engagement in sex–tech behavior. We acknowledge this on pg. xx but also point out that such norms or socio-cultural ideals do not necessarily mean that citizens are not engaging in those taboo or forbidden behaviors (pg. 4): 

“On the other hand, countries and regions differ in their norms and practices, such as holding more conservative views regarding gender roles. While these factors may influence the use and impact of sex-tech, socio-cultural ideals do not necessarily prevent people from engaging in the proscribed behaviors. For example, foundational research by Alfred C. Kinsey revealed considerable same-sex sexual behavior, although these behaviors were illegal at the time [14,15]. Contemporary research has also demonstrated same-sex and premarital sexual behavior in countries where these behaviors are still illegal, such as in Saudi Arabia and Indonesia [16,17]. The nature and extent of such norms and practices on sex–tech behavior are, as yet, unanswered questions.” 

We also acknowledge this again in our Limitations section on pg. 28:

“Countries with higher GII are likely to be more conservative, especially with regards to sexuality. This likely resulted in a selection bias, skewing our sample toward more liberal sexual attitudes and behaviors.”

Third, you pointed out selection bias due to the method of disseminating the survey via the Clue app, newsletter, and social media accounts of Clue and the Kinsey Institute. We acknowledge this in our Limitations section on pp. 27-28:

“…it should be noted that our survey respondents were not a representative sample, and were most likely to have participated in the study after signing up to the newsletter of a sex-positive women’s health app or via the social media accounts of a well-known sex research institute. People who used that particular app or who followed those social media accounts may have had a pre-established interest in sexual health topics greater than that of the general population.”

Fourth, you mentioned that our participants may have been more self-aware of their sexual health than their local peers who did not participate. We agree and now acknowledge this in the Limitations on pg. 28:

“Our participants may also have been more self-aware of their sexual health than a more representative sample, although awareness does not necessarily indicate differences in behavior. Anthropological studies at the community level are needed to answer these questions. Thus, our findings provide a foundation for more detailed exploration in specific contexts.”

Thank you again for your detailed review.

Reviewer #2

Reviewer #2: This is a very fine paper and I don't think any major revisions are necessary. The topic is fairly novel and interesting, uses methods that are fairly standard in the field, and is an easy read. Kudos to the authors for a well done manuscript.

Thank you so much, we appreciate your positivity and the time and effort you contributed to reviewing our paper.

---

## [Editor Report · Decision Letter 1]

19 Aug 2020

Mobile sex-tech apps: How use differs across global areas of high and low gender equality

PONE-D-20-06636R1

Dear Dr. Gesselman,

We’re pleased to inform you that your manuscript has been judged scientifically suitable for publication and will be formally accepted for publication once it meets all outstanding technical requirements.

Kind regards,

Amy Michelle DeBaets, PhD

Academic Editor

PLOS ONE
---

## [Editor Report · Acceptance letter]

24 Aug 2020

PONE-D-20-06636R1 

Mobile sex-tech apps: How use differs across global areas of high and low gender equality 

Dear Dr. Gesselman:

I'm pleased to inform you that your manuscript has been deemed suitable for publication in PLOS ONE. Congratulations! Your manuscript is now with our production department. 

Kind regards, 

on behalf of

Dr. Amy Michelle DeBaets 

Academic Editor

PLOS ONE